# *Super*-Superselective Level VB Neck Dissection for Papillary Thyroid Cancer

**DOI:** 10.3390/cancers17091497

**Published:** 2025-04-29

**Authors:** Dana M. Hartl, Davide Lombardi, Ricard Simo, Radu Mihai, Aleix Rovira, Enyi Ofo, Iain J. Nixon

**Affiliations:** 1Division of Surgery and Anesthesiology, Head and Neck Oncology Service, Thyroid Surgery Unit, Gustave Roussy Cancer Campus Grand Paris, 94800 Villejuif, France; 2Department of Otorhinolaryngology-Head Neck Surgery, University of Brescia, 25121 Brescia, Italy; 3Department of Otorhinolaryngology Head and Neck Surgery, Head, Neck and Thyroid Oncology Unit, Guy’s and St Thomas’ Hospital NHS Foundation Trust, London SE1 7EH, UK; 4Department of Endocrine Surgery, Churchill Cancer Centre, Oxford University Hospitals NHS Foundation Trust, Oxford OX3 9DU, UK; 5Consultant Otorhinolaryngologist Head and Neck Surgeon, Department of Otorhinolaryngology Head and Neck Surgery, St George’s Hospital, London SW17 0QT, UK; 6Department of Otolaryngology Head and Neck Surgery, University of Edinburgh, Edinburgh EH8 9YL, UK

**Keywords:** thyroid cancer, lateral neck dissection, level V, spinal accessory nerve, neuromonitoring

## Abstract

Dissection of neck level VB carries a risk of morbidity related to the dissection of the spinal accessory nerve (SAN), with shoulder pain and functional deficit, even when care is taken to preserve the nerve. A systematic neck dissection of level V is recommended in patients with thyroid cancer classified as cN1b, with clinically apparent lateral neck metastases, even when level V is not involved clinically. As a means to attempt to reduce the risk of SAN morbidity in prophylactic level VB dissection, we propose a technique that limits dissection of the nerve while addressing the lower level of VB.

## 1. Introduction

Therapeutic lateral neck dissection is recommended for papillary thyroid cancer with metastatic lymph nodes detected on palpation or on preoperative imaging (cN1b) [1,2,3]. Modified radical and selective neck dissections involve removing lymphatic tissue in the neck while preserving the vascular, muscular and neural structures, with removal of levels I–V in a modified radical neck dissection (MRND), but removal of fewer levels, according to the site of the primary tumor, in selective neck dissections (SND). Neck levels III and IV are the lateral neck levels most frequently involved in differentiated thyroid cancer, but other neck levels may be involved either clinically or contain occult (microscopic) lymph node metastases [4,5,6]. Lateral neck dissection is indicated for thyroid cancer only when there is preoperative evidence of lymph node metastases in the lateral neck (cN1b) and is not recommended prophylactically in cN0 patients [2]. In patients with macroscopic lymph node metastases diagnosed clinically (cN1b), it is currently recommended to perform a complete neck dissection including levels IIA, III and IV and VB [3].

Level V, or the posterior triangle, divided into two sub-levels, VA (cranial) and VB (caudal), is affected to varying degrees in thyroid cancer. Thus, a systematic dissection of this neck level has been suggested for all patients with cN1b disease, even if level V is not clinically involved by disease [6,7]. The spinal accessory nerve (SAN), the XIth cranial nerve, courses through this region and is vulnerable to post-dissection dysfunction with paresis or paralysis of the trapezius muscle. Even when the SAN is successfully preserved, level V dissection has been related to worse shoulder function [8]. A careful analysis of the risk–benefit ratio of systematic level V neck dissection is needed to optimize functional results for thyroid cancer patients whose prognosis is excellent in most cases. We propose a compromise for systematic level VB dissection in cN1b patients with no clinical evidence of disease in level V: a partial or “*super*-superselective” dissection restricted to below the level of the spinal accessory nerve to address potential occult metastatic lymph nodes in this level while minimizing the functional risk.

## 2. Diagnosis of Clinical Disease in Papillary Thyroid Cancer

Preoperative ultrasound (US) of all neck levels is the recommended primary modality for work-up of papillary thyroid carcinoma [2]. Fine needle aspiration cytology (FNAC) should be performed for suspicious lymph nodes. Suspicious lymph nodes have been defined as harboring cystic areas, peripherally or diffusely increased vascularization, microcalcifications, and/or having a thyroid-like appearance (Figure 1) [9]. Thyroglobulin (Tg) titers should be measured in the needle washout fluid for increased sensitivity in papillary carcinoma [10]. When cytology and/or thyroglobulin levels in the needle washout fluid are in favor of lymph node metastases (LNM), in the setting of first-line therapy a therapeutic neck dissection is recommended [2].

## 3. Terminology of Neck Dissections

Prophylactic, or elective, neck dissection is defined as a neck dissection performed in the absence of clinically detected metastatic lymph nodes (cN0). In papillary thyroid cancer, for the central neck (level VI) it is controversial, but for the lateral neck (I–V) is not recommended [2]. Therapeutic neck dissection is defined as that performed when clinical lymph nodes are present. cN1a refers to clinically detected LNM in the central compartment (level VI) whereas cN1b refers to clinical LNM in one or more lateral compartments (levels I–V) [3,11]. However, in the context of a therapeutic neck dissection (cN1b necks), non-clinically-involved neck levels may be dissected. This is the current recommendation for cN1b papillary thyroid carcinoma, which recommends dissection of ipsilateral level VB prophylactically even in the absence of clinical disease in that level.

The current review will only address this issue in the context of cN1b papillary thyroid carcinoma.

The techniques and extent of neck dissection have evolved since the first reports by Jawdynski 1888 [12,13]. Crile et al. in the early 20th century described the radical neck dissection involving the resection of the sterno-cleido-mastoid muscle (SCM), the SAN and the internal jugular vein, for treatment of head and neck squamous cell carcinoma [14,15]. The technique was later promoted by Martin et al. and became a standard procedure [16]. Due to the morbidity and with increasing data on lymph node involvement in head and neck cancers and thyroid cancers, new techniques evolved to preserve the SCM, the SAN and the internal jugular vein, with the advent of the MRND [17,18,19]. Further refinements led to the concept of selective compartment-oriented neck dissection, which involved removing only the lymphatic structures, fat and connective tissue and preserving even more blood vessels, nerves and muscles, and removing only necessary neck levels according to the location of the primary head and neck tumor [3,20,21,22,23].

A superselective neck dissection is thus defined as the removal of only one or two contiguous neck levels [24]. The concept was developed for treatment of residual disease after chemoradiation therapy for head and neck squamous cell carcinoma, as an alternative to MNRD, and was shown to decrease the functional deficit associated with SAN dysfunction [25]. The concept was later applied to prophylactic treatment of the cN0 neck in various locations of head and neck squamous cell carcinoma as primary therapy and more recently in papillary thyroid carcinoma [23,26,27,28].

The technique that we describe herein is a further reduction of the extent of neck dissection, that we term partial superselective neck dissection or “*super*-superselective neck dissection.” It is the partial, prophylactic, dissection of one neck level, level VB, in an attempt to reduce morbidity. However, contrary to superselective neck dissection in which only one or two complete contiguous neck levels are dissected, the partial dissection of level VB is not applied in isolation, but added to the therapeutic neck dissection of neck levels IIA, III and IV to treat level VB according to current guidelines but reduce morbidity [3]. This concept applies to initial therapy and does not address the issue of recurrent disease, which will not be discussed in the present report.

## 4. Incidence of Level V Involvement in Thyroid Cancer

The reported incidence of level V metastases in thyroid cancer ranges widely from 0% to 53% [6,29,30,31,32]. Level VA involvement in thyroid cancer is rare, however, ranging from 0% to 13% in neck dissection specimens [4,33,34]. Level Vb, is affected in 3.7% to 53% of reported series [4,32,33,34,35,36,37,38].

The main reported risk factor for occult or clinical level VB involvement is the presence of clinically detected lymph nodes in ipsilateral levels II, III and IV [4,6,39,40,41]. Other reported risk factors include a primary tumor size > 2.5 cm [30,42] and contralateral or large (>3 cm) ipsilateral level VI lymph node metastases [5,6,30,43]. Young age, lymphovascular invasion, extrathyroidal extension, tumor multifocality and a BRAF V660E mutation have also been found to be associated with level VB involvement [5,6,30,31,35,36]. The wide reported range of level VB lymph node metastases is due to differences in tumor stages and characteristics, the extent of macroscopic lymph node involvement in level VB and in the other neck levels and whether level VB was dissected prophylactically or due to the presence of known disease.

Given the risk of occult disease in level VB in patients with clinically detected lateral lymph node metastases, the American Thyroid Association consensus statement recommends systematically performing a selective neck dissection of levels IIA, III, IV and VB [3]. However, other authors have suggested that a prophylactic level VB dissection may not be necessary in selected patients with no risk factor for level VB disease, an in particular in the absence of simultaneous level II-III-IV disease [44].

## 5. Definition of Level V Neck Dissection

The techniques and extent of neck dissection have evolved since the first reports by Jawdynski 1888 [12,13]. Crile et al. in the early 20th century described the radical neck dissection involving the resection of the sterno-cleido-mastoid muscle (SCM), the SAN and the internal jugular vein, for treatment of head and neck squamous cell carcinoma [14,15].

The technique was later promoted by Martin et al. and became a standard procedure [16]. Due to the morbidity and with increasing data on lymph node involvement in head and neck cancers and thyroid cancers, new techniques evolved to preserve the SCM, the SAN and the internal jugular vein, with the advent of the MRND [17,18,19]. Further refinements led to the concept of selective compartment-oriented neck dissection, which involved removing only the lymphatic structures, fat and connective tissue, and removing neck levels at risk according to the location of the primary head and neck tumor [3,20,21,22,23]. Neck levels are defined according to anatomic structures, and the neck levels the most frequently involved according to the site of the primary tumor have been identified [45]. The term superselective neck dissection is used to describe neck dissection in which only one or two neck levels are addressed.

Level V is located posteriorly to the posterior border of the sternomastoid muscle, anteriorly to the trapezius muscle, caudal to the skull base at the convergence of these two muscles, and cranial to the clavicle [20]. It is divided into levels VA, superiorly, and VB, inferiorly by a virtual plane at the level of the inferior border of the cricoid cartilage. The SAN and cervical roots comprise the main anatomic structures that need to be dissected and preserved while performing a selective level V neck dissection (Figure 2).

The current technique involves an en bloc dissection of lymph nodes contained in the soft tissue outlined by established anatomical landmarks, with level VB being resected en bloc with the other neck levels. Then, when sending the specimen for final histology, the specimen is either pinned to a support and annotated, or the specimen is divided into sections corresponding to the different neck levels. In both cases, the boundaries between each level are thus relatively subjective, which may lead to discrepancies in the recorded rates of lymph node metastases, particularly between levels IV and VB. This introduces significant bias in the reported rate of involvement of level V lymph nodes.

## 6. Morbidity of Level V Neck Dissection

In the posterior triangle, the SAN enters at the posterior border of the SCM. In anatomic subjects, in has been found to emerge in 90% of subjects an average of 1.4 to 2 cm above the point where the sensory cervical roots cross the posterior border of the muscle (Erb’s point) [46,47]. Kierner et al., however, found that the distance from the clavicle, an average of 8.2 cm, to the point of entry of the nerve was a more reliable landmark [48]. The cervical roots C2 to C4 contribute anatomically and functionally to the SAN in up to 20% of patients, and their injury may also contribute to postoperative functional deficits [48,49,50].

Even in cases where the nerve can be preserved intact without macroscopic damage, dissection of the SAN can cause temporary or permanent nerve dysfunction, either by traction injury, thermal injury by electrocautery devices or by devascularization of the nerve. The data concerning outcomes after dissection of level VB are primarily based on studies of patients treated for head and neck squamous cell carcinoma. Temporary or permanent or paralysis of the trapezius muscle has been reported in 29–39% of patients after neck dissection with preservation of the spinal accessory nerve [51,52,53]. Adding a level VB dissection to a level II-III-IV dissection increased the rate of shoulder dysfunction at one year in the study by Cappiello et al. [54]. A decrease in muscle strength occurred in 20% of patients with a level VB dissection, as compared to no decrease in patients without level VB dissection, but pain was reported in 15% of patients in both groups. Electromyographic abnormalities in the trapezius muscle were detected in 85% of patients in the group with level VB dissection, as compared to 20% of patients in the group with only a level II-III-IV dissection. Dissection of level VB may also increase the rate of chronic pain after neck dissection. In the study by Van Wilgen et al., 51% of patients treated for chronic pain 3 or more years following neck dissection had an objective decrease in muscle strength and mobility in the trapezius muscle, and conversely, 69% of patients with dysfunction of the trapezius muscle had chronic pain [55]. Preserving the cervical roots decreases the incidence of chronic pain, with a decrease from 72% to 37% in the study by Gane et al. [53].

This functional risk is the main issue when evaluating the risk–benefit ratio of performing level VB neck dissection, particularly in thyroid cancer which in many cases affects young patients with a long life expectancy [56,57].

## 7. Concept of *Super*-Superselective Level VB/Extended Level IV and Surgical Technique

Contrary to level IIB dissection, which is addressed by an approach medial to the sternomastoid muscle, level VB may be dissected either from a lateral approach, dissecting the skin posterior to the sternomastoid muscle, or from a medial approach, by dissecting behind the posterior border of the sternomastoid muscle.

To perform a complete selective level VB neck dissection, in both approaches, the dissection must encompass the area from the virtual level of the cricoid cartilage superiorly, down to the clavicle inferiorly and posteriorly to the trapezius muscle, dissecting the spinal accessory nerve and cervical roots contained in that space.

In thyroid cancer, for patients with lateral lymphadenopathy in levels II, III and IV, but in the absence of radiologically-proven lymph node metastases in level VB, given the relatively high rate of occult disease as demonstrated above, it is recommended to perform an elective or prophylactic neck dissection of level VB in these patients, with the risk of spinal accessory nerve injury [35]. However, a surgical compromise can be reached in the form of what may be termed a “*super*-superselective” level VB neck dissection or an “extended level IV” neck dissection. This can be performed by limiting the dissection to the lower portion of level VB, without extensive dissection of the spinal accessory nerve. The dissection is caudal to the lowest cervical roots and Erb’s point. This addresses the low-lying lymph nodes in the region, which are the most likely to be affected by thyroid carcinoma.

The SAN can be identified by neurostimulation, and traced posteriorly, with the neck dissection proceeding caudally to the nerve, thus avoiding direct dissection of the nerve. Identification and preservation of functional branches of the cervical roots that contribute to the SAN is also facilitated with the use of intraoperative nerve stimulation [58]. To facilitate this dissection, the surgeon may stand on the opposite side of the neck dissection, to improve visibility of the area under the skin flap and/or SCM. The medial approach has the advantage of not requiring the dissection of a posterior skin flap, which may potentially cause nerve stretching, devascularization or inadvertent injury. The SAN can be mapped using neuromonitoring while still surrounded by fatty tissues and its blood supply, therefore minimizing the risk of neural injury (Figure 3).

Rates of intra-operative nerve monitoring in thyroid surgery are increasing globally [59,60,61]. Although the main use of this technology is monitoring the recurrent laryngeal nerve, it can also be applied to monitoring of the SAN or other motor nerves [62,63,64]. Either with visualization of muscle contraction or with the use of electrodes, nerve stimulator technology can be used to confidently identify the SAN as it passes behind the sternocleidomastoid muscle in the posterior triangle.

This concept of subdividing level VB and individualizing the lower part of this neck level is already in use by radiation oncologists who have divided level V into three levels: VA, VB and VC [65]. Level VC was initially described in the setting of nasopharyngeal carcinoma and is defined as being limited by the transverse cervical vessels superiorly, the trapezius muscle posteriorly, the clavicle inferiorly and level IV anteriorly. It is, in fact, the lower portion of the surgically defined level VB.

## 8. Conclusions

Metastatic lymph nodes in level VB are found to varying degrees in thyroid cancer. The main risk factor for overt or occult disease in level VB is the simultaneous involvement of neck levels II, III and IV, and routine dissection of level VB is recommended in patients with macroscopic lymph node metastases in the lateral neck. Systematic and complete dissection of level VB exposes patients to temporary or permanent shoulder dysfunction and chronic pain, even when care is taken to preserve the spinal accessory nerve and cervical roots. Our practice is to electively dissect the lower portion of level VB, when this level is not clinically involved, as a “*super*-superselective” level VB dissection or an “extended level IV” dissection, avoiding extensive dissection of the spinal accessory nerve and cervical roots in the posterior triangle (Figure 4). Although not mandatory, intraoperative neuromonitoring may facilitate identification of the spinal accessory nerve and functionally significant cervical roots, to decrease the extent of nerve dissection and reduce the risk of a postoperative functional deficit.

## 9. Future Directions

The main weakness of the present report is the lack of data supporting the practice of a prophylactic dissection of level VB in patients with lateral LNM involving levels II, III and/or IV. Despite current guidelines, the optimal extent of lateral neck dissection in patients with clinically detected lateral lymph node metastases remains controversial. The existing evidence is based on retrospective reports of heterogeneous cohorts. We describe our current practice, with the opinion that it improves outcomes as compared to a complete level VB dissection in terms of function, but we have no evidence from prospective clinical trials to prove that this is true or to prove that this type of partial prophylactic neck dissection of level VB improves oncologic outcomes in patients who have metastatic nodes in other neck levels.

Prospective randomized trials provide the highest level of evidence. Ideally, the question of prophylactic level VB neck dissection in cN1b thyroid cancer patients should be resolved with a prospective trial comparing therapeutic neck dissection with prophylactic level VB dissection versus partial level VB dissection (*super*-superselective level VB) versus no level VB dissection at all in patients with only clinically detectable nodes in other neck levels.

With the existing evidence, an expert consensus or new clinical practice guidelines specifying the indications for prophylactic level VB dissection—and when a level VB dissection may be avoided –could aid in reducing the existing controversy and aid clinicians in decision-making. Until there is higher-level evidence or a consensus, a prophylactic or *super*-superselective dissection of the lower part of level VB in cN1b patients may provide a compromise between a full level VB dissection with risk of SAN morbidity and no level VB dissection at all.

The rare cases of thyroid cancer with preoperatively confirmed macroscopic metastatic disease in level VB will continue to require, however, a comprehensive selective neck dissection of levels IIA-IV including VB with careful dissection and preservation of the SAN.

## Figures and Tables

**Figure 1 cancers-17-01497-f001:**
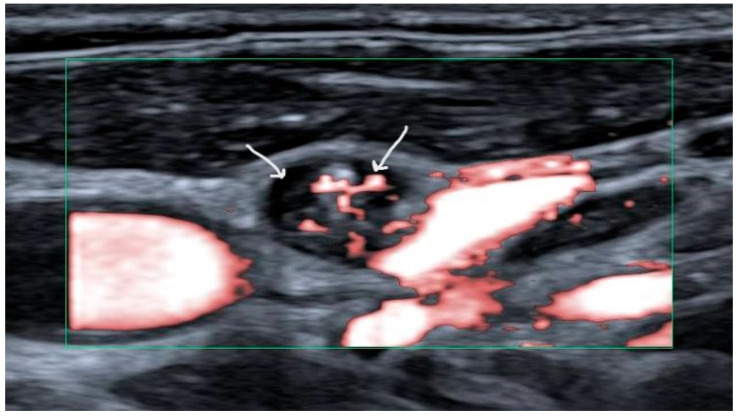
Ultrasound image showing a suspicious lymph node with a cystic area (left arrow) and hypervascularization (right arrow).

**Figure 2 cancers-17-01497-f002:**
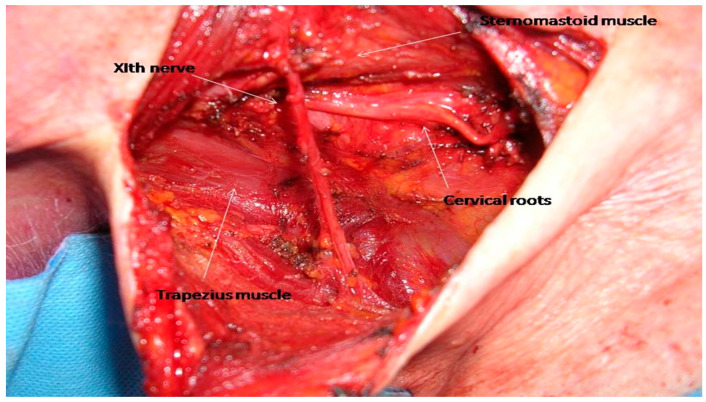
Complete dissection of levels VA and VA with dissection of the right SAN.

**Figure 3 cancers-17-01497-f003:**
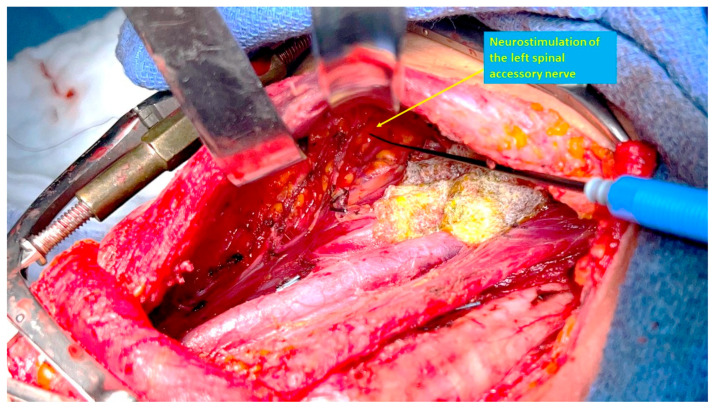
Neurostimulation of the left spinal accessory nerve in an extended level IV/*super*-superselective level VB neck dissection. Lymphostatic gauze has been placed on the lower part of the internal jugular vein at the level of the thoracic duct just above the clavicle.

**Figure 4 cancers-17-01497-f004:**
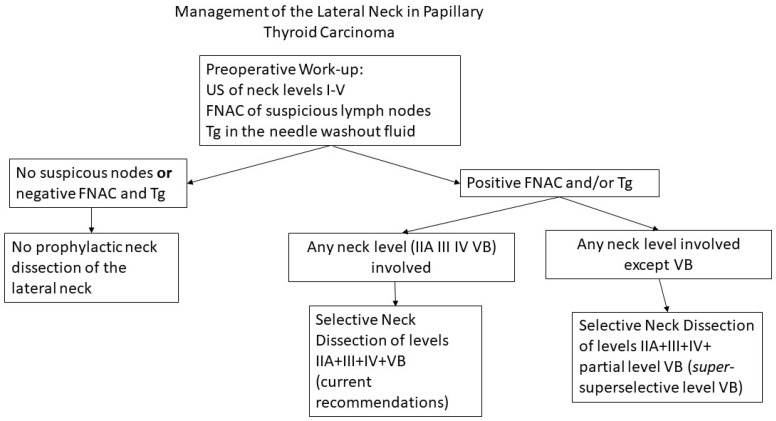
Proposed algorithm for management of the lateral neck in papillary thyroid carcinoma.

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
