# Peer review of "Super-Superselective Level VB Neck Dissection for Papillary Thyroid Cancer"

_cancers, 2025, doi:10.3390/cancers17091497_

Round 1

Reviewer 1 Report

Comments and Suggestions for Authors

Dear editor,

Dear author(s),

It is my opinion that the submitted review manuscript is a well-written narrative review, with the focus on surgical approach to neck metastatic lymph nodes in thyroid cancer patients.

Super-selective neck dissection consists of surgical ressection of the fibroareolar tissue contents of two or less contiguous neck levels, as defined by Verma and Chen in a 2020 review addressing the indications and outcomes of superselective neck dissection (Verma A, Chen AY. Laryngoscope Investig Otolaryngol. 2020 Jun 25;5(4):672-676).  Super-selective neck dissection has been applied to the treatment of the necks in N0 stage or after chemoradiation.

Herein, the authors argue in favor of extending lateral neck dissection to level VB and propose an approach to avoid surgical intercurrences.

The authors acknowledge that the issue of extending lateral neck dissection to level V is highly controversial and that is, in my perspective, one of the main reasons for the pertinence for publishing the present manuscript. Another reason is addressing the major intercurrences and morbidities secondary to level V neck dissection, namely spinal accessory nerve dysfunction, by proposing how to overcome / avoid it and early identify it.

Nevertheless, there are some issues that still need to be addressed, which are listed below:

1) Well-differentiated thyroid carcinomas are commonly divided in medullary, papillary and follicular thyroid carcinomas.

However, in the present manuscript the authors do not distinguish between different types of differentiated thyroid tumors. Moreover, the authors do not mention to which subtype of differentiated thyroid cancer is addressed by their review or if the super-selective level VB neck dissection approach is applicable to all the thyroid carcinoma subtypes.

Therefore, authors should clarify within the manuscript text the following questions:

  1. i) are there different indications for super-selective level VB neck dissection for each of differentiated thyroid carcinomas?
  2. ii) are there any technical differences, even if subtle, when approaching papillary, follicular or medullary thyroid carcinomas?

2) From the different types of differentiated thyroid carcinomas, papillary thyroid carcinoma is the most prevalent (about 70-80% of thyroid carcinomas) and the least aggressive, whereas follicular thyroid carcinoma is a more aggressive subtype and accounts for about 10 % of thyroid malignancies (Amaral M, Afonso RA, Gaspar MM, Reis CP. Anaplastic thyroid cancer: How far can we go? EXCLI J. 2020 Jun 15;19:800-812.).

Accordingly, most of the studies addressing the super-selective neck dissection surgical approach in thyroid tumors refer specifically to papillary thyroid tumors, including the report from Liang and co-workers published this year (Liang S, Bellamkonda N, Tullis B, Hunt JP. Superselective Versus Selective Neck Dissection in the Treatment of Papillary Thyroid Carcinoma. Ann Otol Rhinol Laryngol. 2025 Apr;134(4):254-258).

Is the present report restricted to papillary thyroid carcinoma? Such issue should be clarified.

3) In 2014, Kim et al (Kim H, Jin YJ, Cha W, Jeong WJ, Ahn SH. Feasibility of super-selective neck dissection for indeterminate lateral neck nodes in papillary thyroid carcinoma. Head Neck. 2014 Apr;36(4):487-91) and in 2022, Song et al (reference 22) suggested that lateral neck dissection including levels III to IV should be considered prophylactically may be considered in patients with papillary thyroid canrcinoma, even when no lymph node invasion was identified.

However, in a study published in 2017 by An C et al , it is suggested that level V selective dissection is not mandatory since only 1.4% occult lymph node metastasis were observed in patients presenting lymph node metastases in levels II-III (An C, Zhang X, Wang S, Zhang Z, Yin Y, Xu Z, Tang P, Li Z. Efficacy of Superselective Neck Dissection in Detecting Metastasis in Patients with cN0 Papillary Thyroid Carcinoma at High Risk of Lateral Neck Metastasis. Med Sci Monit. 2017 May 4;23:2118-2126). These authors further suggest that super-selective neck dissections of levels II, III, and IV may be effective for early diagnosis of lateral neck metastases of papillary thyroid carcinoma, but level V dissection could be omitted.

In the light of this debate, do the authors consider that the surgical approach should also include ipsilateral VB level dissection, despite no macroscopic lymph node metastases are identified?

Both in section 5 (lines 154-158) and in their conclusions (page 4, lines 192-195), the authors seem to suggest such approach even in the absenc of lymph node metastases in level VB. If so it would be advisable to provide evidence (i.e., previous studies) showing a clear risk-benefit assessment lining towards VB level dissection.

4) Additionally, Kupferman et al (reference 21) previously published a study in which they observed that in 57 % cases contralateral neck dissections also revealed to be positive for metastases. Based on this, what would be the authors perspective on also performing contralateral (prophylactic) neck dissection? This topic could be discussed in either sections 5 or 6.

5) The authors propose that one can minimize secondary lesion of spinal acessory nerve by limiting the dissection to the lower portion of level VB, with the dissection being caudal to the lowest cervical roots and Erb’s point, and additionally applying neurostimulation of the spinal accessory nerve.

However, the authors do not present consistent statistics on the change of outcome by applying the suggested surgical approach and on the incidence of spinal acessory nerve lesion. This issue should be addressed within section 5.

6) Finally, the references used within these commentaries are relatively recent (some from 2025) and it would be beneficial to the current manuscript to include them.

Author Response

Dear Editor, Dear Reviewers,

We thank you for the time and effort that you have put into reviewing our manuscript. Thank you for the thoughtful questions and recommendations that have improved our work. Below are our responses to each specific issue raised. We have reorganized the manuscript and added more content and references, as per the reviewers’ suggestions. Given the number of changes, we have not highlighted them in the text, but the changes can be tracked from the Word version of the revised manuscript that we are sending.

Thank you again for your kind consideration.

Sincerely,

Dana Hartl, corresponding author

RESPONSES TO REVIEWERS:

Reviewer 1

Dear author(s),

It is my opinion that the submitted review manuscript is a well-written narrative review, with the focus on surgical approach to neck metastatic lymph nodes in thyroid cancer patients.

Super-selective neck dissection consists of surgical ressection of the fibroareolar tissue contents of two or less contiguous neck levels, as defined by Verma and Chen in a 2020 review addressing the indications and outcomes of superselective neck dissection (Verma A, Chen AY. Laryngoscope Investig Otolaryngol. 2020 Jun 25;5(4):672-676).  Super-selective neck dissection has been applied to the treatment of the necks in N0 stage or after chemoradiation.

Thank you for your remark and precision. We have added this definition, the reference and the history of the development of the concept of superselective neck dissection in the treatment of head and neck squamous cell carcinoma in a new chapter (new chapter 3) under the heading of “terminology for neck dissection.” We have added other references as well. This will make the concept much clearer.

Suarez C, Rodrigo JP, Robbins KT, et al. Superselective neck dissection: rationale, indications, and results. Eur Arch Otorhinolaryngol. 2013;280(11):2815‐2821.

Avanti Verma 1,, Amy Y Chen

Laryngoscope Investig Otolaryngol

. 2020 Jun 25;5(4):672–676. doi: 10.1002/lio2.421 PMCID: PMC7444773  PMID: 32864437

Herein, the authors argue in favor of extending lateral neck dissection to level VB and propose an approach to avoid surgical intercurrences.

The authors acknowledge that the issue of extending lateral neck dissection to level V is highly controversial and that is, in my perspective, one of the main reasons for the pertinence for publishing the present manuscript. Another reason is addressing the major intercurrences and morbidities secondary to level V neck dissection, namely spinal accessory nerve dysfunction, by proposing how to overcome / avoid it and early identify it.

Nevertheless, there are some issues that still need to be addressed, which are listed below:

1) Well-differentiated thyroid carcinomas are commonly divided in medullary, papillary and follicular thyroid carcinomas.

However, in the present manuscript the authors do not distinguish between different types of differentiated thyroid tumors. Moreover, the authors do not mention to which subtype of differentiated thyroid cancer is addressed by their review or if the super-selective level VB neck dissection approach is applicable to all the thyroid carcinoma subtypes.

Therefore, authors should clarify within the manuscript text the following questions:

  1. i) are there different indications for super-selective level VB neck dissection for each of differentiated thyroid carcinomas?
  2. ii) are there any technical differences, even if subtle, when approaching papillary, follicular or medullary thyroid carcinomas?

We have modified the manuscript to only address papillary thyroid carcinoma. There is very little data specifically regarding level VB medullary or follicular carcinoma.

2) From the different types of differentiated thyroid carcinomas, papillary thyroid carcinoma is the most prevalent (about 70-80% of thyroid carcinomas) and the least aggressive, whereas follicular thyroid carcinoma is a more aggressive subtype and accounts for about 10 % of thyroid malignancies (Amaral M, Afonso RA, Gaspar MM, Reis CP. Anaplastic thyroid cancer: How far can we go? EXCLI J. 2020 Jun 15;19:800-812.).

Accordingly, most of the studies addressing the super-selective neck dissection surgical approach in thyroid tumors refer specifically to papillary thyroid tumors, including the report from Liang and co-workers published this year (Liang S, Bellamkonda N, Tullis B, Hunt JP. Superselective Versus Selective Neck Dissection in the Treatment of Papillary Thyroid Carcinoma. Ann Otol Rhinol Laryngol. 2025 Apr;134(4):254-258).

Is the present report restricted to papillary thyroid carcinoma? Such issue should be clarified.

Thank you for the reference that we have added in the new section 3. Yes, we have restricted our report to papillary thyroid carcinoma. There is very little data specifically regarding level VB in medullary or follicular carcinoma and no data in anaplastic thyroid carcinoma.

3) In 2014, Kim et al (Kim H, Jin YJ, Cha W, Jeong WJ, Ahn SH. Feasibility of super-selective neck dissection for indeterminate lateral neck nodes in papillary thyroid carcinoma. Head Neck. 2014 Apr;36(4):487-91) and in 2022, Song et al (reference 22) suggested that lateral neck dissection including levels III to IV should be considered prophylactically may be considered in patients with papillary thyroid carcinoma, even when no lymph node invasion was identified.

However, in a study published in 2017 by An C et al , it is suggested that level V selective dissection is not mandatory since only 1.4% occult lymph node metastasis were observed in patients presenting lymph node metastases in levels II-III (An C, Zhang X, Wang S, Zhang Z, Yin Y, Xu Z, Tang P, Li Z. Efficacy of Superselective Neck Dissection in Detecting Metastasis in Patients with cN0 Papillary Thyroid Carcinoma at High Risk of Lateral Neck Metastasis. Med Sci Monit. 2017 May 4;23:2118-2126). These authors further suggest that super-selective neck dissections of levels II, III, and IV may be effective for early diagnosis of lateral neck metastases of papillary thyroid carcinoma, but level V dissection could be omitted.

In the light of this debate, do the authors consider that the surgical approach should also include ipsilateral VB level dissection, despite no macroscopic lymph node metastases are identified?

Thank you for this remark. We were not clear about the indication. We have tried to make the issue clearer, in a new chapter 3 “terminology of neck dissections.” Prophylactic dissection of level VB, or other lateral neck levels, is not recommended in cN0 papillary thyroid carcinoma. Dissecting level VB prophylactically--that is, when not clinically involved—is recommended, however, for cN1b patients. There is indeed a wide range of reported incidences of occult level VB metastases in papillary thyroid carcinoma, possibly due to hetereogeneous cohorts. We have added this information in the new section. The study by An et al. addressed patients with cN0 carcinoma whereas in our report we address the issue of cN1B patients with clinically proven lateral neck metastases. 

Both in section 5 (lines 154-158) and in their conclusions (page 4, lines 192-195), the authors seem to suggest such approach even in the absence of lymph node metastases in level VB. If so it would be advisable to provide evidence (i.e., previous studies) showing a clear risk-benefit assessment lining towards VB level dissection.

There is currently no high-level evidence in the literature showing that this approach improves outcomes. The current guidelines are based on retrospective data of heterogeneous cohorts reporting varying incidences of occult lymph node metastases in level VB. We have added more data on the incidence of subclinical (occult) lymph node metastases in level VB in cN1b patients in new chapter 4.

4) Additionally, Kupferman et al (reference 21) previously published a study in which they observed that in 57 % cases contralateral neck dissections also revealed to be positive for metastases. Based on this, what would be the authors perspective on also performing contralateral (prophylactic) neck dissection? This topic could be discussed in either sections 5 or 6.

We have added definitions and the current recommendations in new chapter 3 in order to be more precise. In Kupferman et al., 13 patients underwent contralateral neck dissection for clinical nodes in the contralateral neck, with 57% being positive in level V, which is comparable to the 53% on the ipsilateral side. A prophylactic contralateral neck dissection is not currently recommended, but a therapeutic neck dissection is recommended in case of clinically detected lymph node metastases.

5) The authors propose that one can minimize secondary lesion of spinal acessory nerve by limiting the dissection to the lower portion of level VB, with the dissection being caudal to the lowest cervical roots and Erb’s point, and additionally applying neurostimulation of the spinal accessory nerve.

However, the authors do not present consistent statistics on the change of outcome by applying the suggested surgical approach and on the incidence of spinal acessory nerve lesion. This issue should be addressed within section 5.

Thank you for pointing out the main weakness in our review. We describe our current practice, with the expert opinion that it improves outcomes as compared to a complete level VB dissection, but we have no evidence from prospective clinical trials to prove that this is true (and not just our impression) or to prove that this type of partial prophylactic neck dissection of level VB improves oncologic outcomes in patients who have metastatic nodes in other neck levels. We have added this in under the heading “future directions” (new chapter 9). However, existing guidelines are not based on high-level evidence either, only retrospective data and expert opinion.

6) Finally, the references used within these commentaries are relatively recent (some from 2025) and it would be beneficial to the current manuscript to include them.

We thank the reviewer for such a thorough review and for the pertinent references that we have gratefully added.

Reviewer 2 Report

Comments and Suggestions for Authors

Hartl D et al, remarkably conduct a systematic review on the Super-super selective Level VB Neck Dissection for Differentiated Thyroid Cancer

The review article is exceptionally well written and very interesting for clinicians working on this topic.

I have only some minor comments.

I would suggest the authors to add a short paragraph on thyroid ultrasound evidence of pre-operatory metastatic lymph nodes.

Some pictures are also needed.

I think that this point might add a value on the two different situations described: the super-selective level VB neck dissection or the extended level IV neck dissection.

Finally, an exhaustive algorithm about a proposal on the indications for prophylactic level VB dissection should be inserted in the paper.

Author Response

Dear Editor, Dear Reviewers,

We thank you for the time and effort that you have put into reviewing our manuscript. Thank you for the thoughtful questions and recommendations that have improved our work. Below are our responses to each specific issue raised. We have reorganized the manuscript and added more content and references, as per the reviewers’ suggestions. Given the number of changes, we have not highlighted them in the text, but the changes can be tracked from the Word version of the revised manuscript that we are sending.

Thank you again for your kind consideration.

Sincerely,

Dana Hartl, corresponding author

RESPONSES TO REVIEWERS:

Reviewer 2

The review article is exceptionally well written and very interesting for clinicians working on this topic.

I have only some minor comments.

I would suggest the authors to add a short paragraph on thyroid ultrasound evidence of pre-operatory metastatic lymph nodes.

Thank you for your remark which situates the issue much more clearly. We have added a new section 2 to clarify how clinical lymph node metastases are diagnosed.

Some pictures are also needed.

We have added an ultrasound image (new figure 1).

I think that this point might add a value on the two different situations described: the super-selective level VB neck dissection or the extended level IV neck dissection.

For clarity we have deleted the term “extended level IV” and added “partial” level VB.

Finally, an exhaustive algorithm about a proposal on the indications for prophylactic level VB dissection should be inserted in the paper.

We have added an algorithm (new figure 4)

Reviewer 3 Report

Comments and Suggestions for Authors

Great review and very kindly new operative design.
Two suggestions:
1) make an anatomical illustration to clearly draw the extent of super-super selective Lymph node
2)in the future direction: had better emphasize the preoperative echo ,cytology ,and tissue Tg for lymph node evaluations 

Author Response

Dear Editor, Dear Reviewers,

We thank you for the time and effort that you have put into reviewing our manuscript. Thank you for the thoughtful questions and recommendations that have improved our work. Below are our responses to each specific issue raised. We have reorganized the manuscript and added more content and references, as per the reviewers’ suggestions. Given the number of changes, we have not highlighted them in the text, but the changes can be tracked from the Word version of the revised manuscript that we are sending.

Thank you again for your kind consideration.

Sincerely,

Dana Hartl, corresponding author

RESPONSES TO REVIEWERS:

Reviewer 3

Great review and very kindly new operative design.
Two suggestions:
1) make an anatomical illustration to clearly draw the extent of super-super selective Lymph node

Thank you for the excellent idea. Unfortunately we do not have access to an anatomical illustrator to provide a good-quality illustration.2)in the future direction: had better emphasize the preoperative echo ,cytology ,and tissue Tg for lymph node evaluations 

Thank you for the suggestion that aids in clarifying the issue. We have added a chapter regarding preoperative lymph node evaluation with references (new chapter 2).
